# Enantioselective construction of *ortho*-sulfur- or nitrogen-substituted axially chiral biaryls and asymmetric synthesis of isoplagiochin D

He Yang[1] & Wenjun Tang [1,2] ✉

Axially chiral biaryl motifs possessing *ortho*-heteroatom-substituted functionalities exist widely in the structures of natural products and have served as foundation for constructing prominent chiral organocatalysts, ligands, functional materials, and even bioactive molecules. However, a general and enantioselective synthesis of such chiral structures with high synthetic value is rare. Taking advantage of the BaryPhos-facilitated asymmetric Suzuki-Miyaura cross-coupling, we have established a general, efficient and enantioselective construction of the *ortho* sulfur- or nitrogen-substituted axially chiral biaryls. The protocol shows excellent compatibility to various functional groups and structural features, delivering chiral biaryl structures with *ortho*-sulfonyl groups or with *ortho*-nitro groups at a broad range of molecular diversity and complexity. The immobilization of BaryPhos on polyethylene glycol (PEG) support has enabled homogeneous enantioselective cross-coupling in aqueous media and the palladium catalyst recycling for multiple times. The method has enabled a concise 10-step asymmetric synthesis of isoplagiochin D as well as the construction of chiroptical molecules with circularly polarized luminescence (CPL) properties.

Axially chiral biaryl motifs exist in numerous natural products and endue these naturally occurring entities with different structural features and biological activities[1–3]. In medicinal chemistry, bioactive synthetic compounds based on axially chiral biaryl framework have been emerging as a promising class of drug candidates[4,5]. The chiral biaryl scaffolds also set up the foundation for chiral ligands/catalysts development, as exemplified by BINOL, BINAP, and related derivatives which have shown broad application in asymmetric catalysis[6,7]. Among chiral structures of this type, the *ortho* S- or N-substituted chiral biaryls are important and appealing, comprising a substantial proportion in alkaloids (Fig. 1a)[8,9]. In addition, a series of chiral catalysts have been developed based on axially chiral biaryl skeletons with S- or N-substitution at the *ortho* position of the stereogenic biaryl axis, such as the chiral amino alcohol NOBIN and BINOL-derived disulfonimide catalysts[10–12]. In pharmaceutical industry, aryl sulfonyl group-containing compounds, such as sulfonamides, sulfones, and sulfonate, often display various biological activities and belong to a leading class of therapeutic agents. Nevertheless, most of these pharmaceutical compounds to date are achiral or with only central chirality[13]. Drug molecules bearing *ortho*-sulfonyl-substituted axially chiral biaryl scaffolds are rarely explored, but should be highly valuable since the three-dimensional topology exerted by axial chirality would impact their specificity and efficiency in protein binding[4,14]. Therefore, the development of practical and enantioselective synthetic method of

[1]State Key Laboratory of Bio-Organic and Natural Products Chemistry, Center for Excellence in Molecular Synthesis, Shanghai Institute of Organic Chemistry, University of Chinese Academy of Sciences, 345 Lingling Road, Shanghai 200032, China. [2]School of Chemistry and Materials Science, Hangzhou Institute for Advanced Study, University of Chinese Academy of Sciences, 1 Sub-lane Xiangshan, Hangzhou 310024, China. ✉e-mail: tangwenjun@sioc.ac.cn

**a. Axially chiral molecules with *ortho N-* or *S*-functionalization and prevalence of achiral aryl sulfonyl-containing drugs**

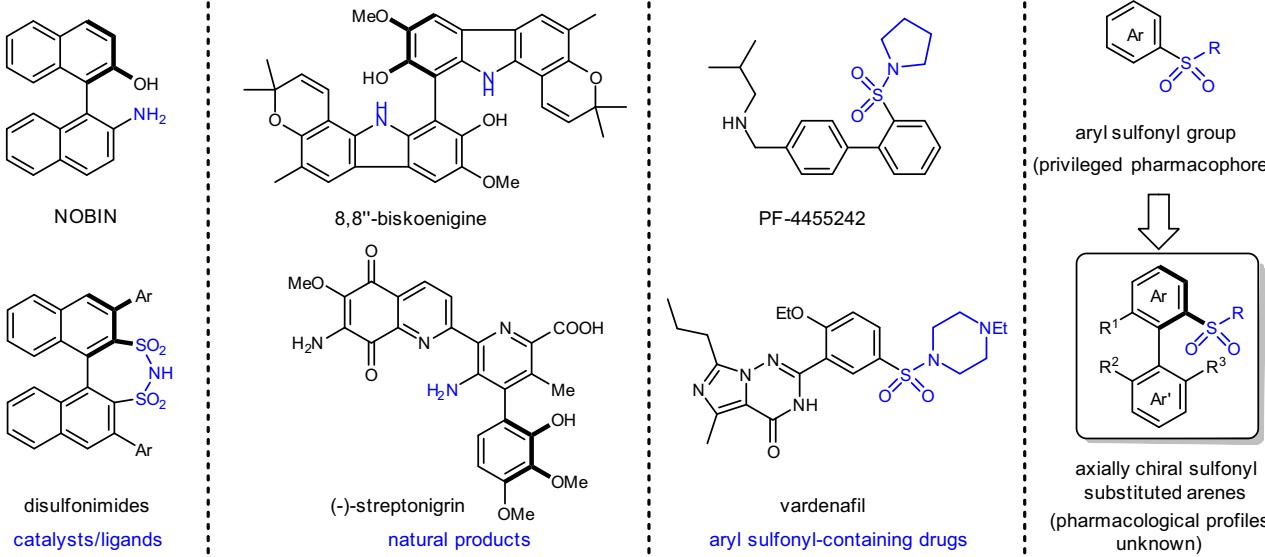

NOBIN

8,8''-biskoenigine

PF-4455242

aryl sulfonyl group
(privileged pharmacophore)

disulfonimides
catalysts/ligands

(-)-streptonigrin
natural products

vardenafil
aryl sulfonyl-containing drugs

axially chiral sulfonyl
substituted arenes
(pharmacological profiles
unknown)

**b. Types of *ortho*-functionalities in chiral biaryls by enantioselective Suzuki-Miyaura cross-coupling**

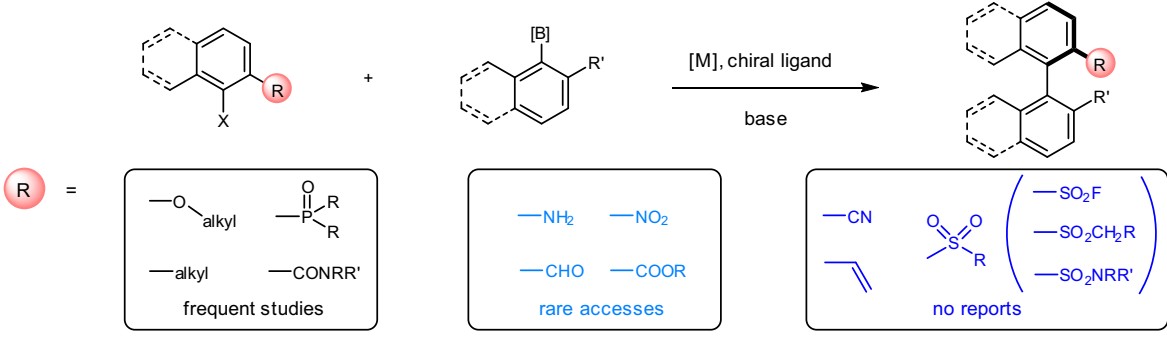

[M], chiral ligand

base

R = 

—O—alkyl    P(=O)R R

—alkyl    —CONRR'

frequent studies

—NH₂    —NO₂

—CHO    —COOR

rare accesses

—CN    S(=O)(=O)R

—SO₂F

—SO₂CH₂R

—SO₂NRR'

no reports

**c. Synthesis of axially chiral *ortho* S- or N-substituted biaryls via cross-coupling (this work)**

- ambient conditions
- recyclable catalyst
- drug relevant products
- versatile post-functionalizations

R = NO₂ or SO₂R'

Pd   BaryPhos

(*S,S*)-BaryPhos

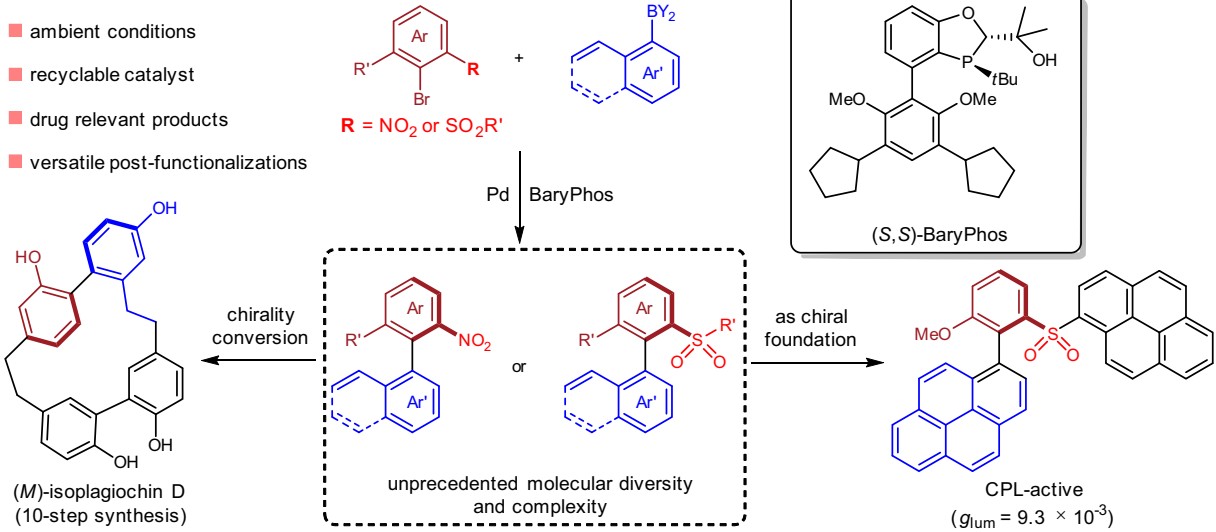

chirality
conversion

(*M*)-isoplagiochin D
(10-step synthesis)

unprecedented molecular diversity
and complexity

as chiral
foundation

CPL-active
($g_{lum} = 9.3 \times 10^{-3}$)

**Fig. 1 | *ortho* S- or N-substituted axially chiral biaryls. a** The prevalence of *ortho* S- or N-substituted axially chiral biaryls and relavent achiral aryl sulfonyl-containing drugs. **b** Limited accessibility of chiral biaryls with *ortho* functionalities by enantioselective Suzuki-Miyaura cross-coupling. **c** Synthesis of axially chiral *ortho* S- or N-substituted biaryls via cross-coupling.

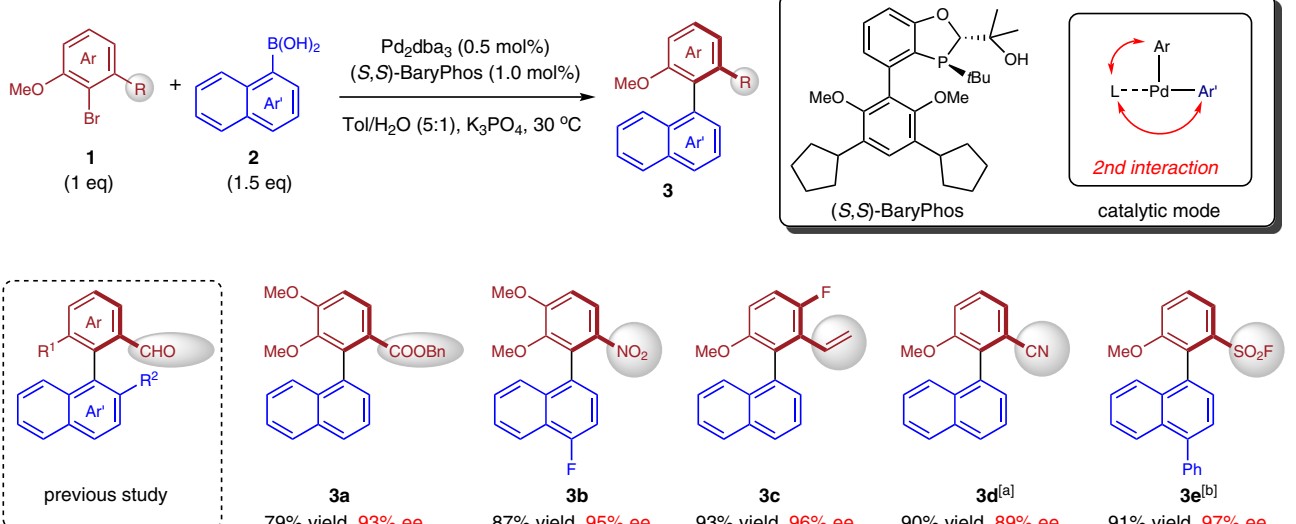

**Fig. 2 | The generality of BaryPhos-facilitated enantioselective cross-coupling.**
Reaction conditions: **1** (0.20 mmol), **2** (0.30 mmol), Pd₂(dba)₃ (0.5 mol%), (*S*,*S*)-BaryPhos (1.0 mol%), K₃PO₄ (0.60 mmol), Tol/H₂O (5 :1), 30 °C, 15 h; the yields refer to isolated yields; the *ee* value was determined by HPLC. [a] Thermal racemization barrier $\Delta G^{\ddagger}$ = 27.6 kcal/mol (Supplementary Fig. 3a). [b] The reaction was conducted at 35 °C in DCE/H₂O (5:1), thermal racemization barrier $\Delta G^{\ddagger}$ = 29.0 kcal/mol (Supplementary Fig. 3b).

*ortho*-sulfonyl-substituted chiral biaryls would certainly enhance their accessibility to medicinal chemists and facilitate the illumination of their biological profiles.

The asymmetric Suzuki-Miyaura cross-coupling has been a highly pursued method for the synthesis of axially chiral biaryl compounds. The last two decades have witnessed appreciable progress in this area, assisted by the development of a variety of prominent chiral ligands[15,16]. Despite the advancement, the employment of enantioselective Suzuki-Miyaura coupling in the synthesis of axially chiral natural products and high value-added molecules remains a challenging task and only a limited number of application have been reported to date[17–21]. In addition, synthetic and medicinal chemists nowadays still suffer from the situation of few readily available, practical and robust enantioselective cross-coupling protocols for the efficient synthesis of highly functionalized bioactive biaryl intermediates with axial chirality. The reported asymmetric Suzuki-Miyaura coupling methodologies are mostly restricted to the preparation of axially chiral biaryl products with *ortho* alkoxy, phosphonyl, alkyl, or amide groups. The *ortho* aldehyde-, ester-, amino-, and nitro-functionalized chiral biaryls which serve as precursors of chiral natural products and catalysts, have been seldom synthesized through this asymmetric cross-coupling method (Fig. 1b)[22–24]. Moreover, the biologically relevant *ortho* cyano-, alkenyl-, and sulfonyl-substituted axially chiral biaryls structures have never been synthesized via enantioselective Suzuki-Miyaura cross-coupling methods[25–29].

Appealed by their remarkable synthetic utilities and potential biomedical applications, we herein report the enantioselective synthesis of *ortho* sulfonyl-substituted axially chiral biaryl compounds and the efficient construction of *ortho* nitro-substituted axially chiral biaryls with unprecedented molecular diversity and complexity via asymmetric Suzuki-Miyaura coupling (Fig. 1c). The post-transformations of these highly functionalized chiral structures have enabled a concise asymmetric synthesis of isoplagiochin D, a highly strained macrocyclic bis(bibenzyls) natural product, and CPL-active chiroptical molecules.

## Results and discussion

### The generality of BaryPhos-mediated enantioselective cross-coupling

Noncovalent interactions between chiral ligands and substrates have been shown to be essential for catalytic asymmetric Suzuki-Miyaura cross-coupling[30,31]. Taking advantage of this tactic, we previously established the efficient synthesis of *ortho*-formyl-substituted axially chiral biaryls using BaryPhos, a P-chiral ligand containing a tertiary alcohol moiety as the hydrogen bond donor which could engage in hydrogen bonding with the CHO group of substrate and promote effective enantiocontrol in cross-coupling[18]. Structurally, substrates with functional groups bearing similar properties in charge distribution to aldehyde were presumed to comply with this catalytic mode. As shown in Fig. 2, a systematic evaluation of aryl bromide substrates carrying varied substituents *ortho* to the Br group demonstrated the feasibility of this concept and the rarely accessed *ortho*-ester (**3a**) and -nitro (**3b**) substituted axially chiral biaryls were synthesized in high yields and enantioselectivities from the coupling between the corresponding aryl bromides and boronic acids. This coupling protocol also enabled the highly enantioselective synthesis of axially chiral biaryl compounds possessing *ortho*-alkenyl (**3c**), cyano (**3d**), and sulfonyl functionalities (**3e**), demonstrating the generality and prominent enantio-induction ability of BaryPhos (see Supplementary Table 2 for ligands comparison).

### Enantioselective synthesis of *ortho* sulfonyl-substituted axially chiral biaryls

The *ortho* S-substituted axially chiral biaryl compounds possess significant application potential in drug discovery and material science. However, their asymmetric synthesis remains a challenging task. Pioneering work from Colobert and co-workers enabled the efficient synthesis of *ortho* sulfinyl-substituted axially chiral biaryls via a sulfoxide-directed asymmetric C-H functionalization strategy[28,29]. Zhao and co-workers realized an electrophilic carbothiolation of alkynes for the synthesis of chiral sulfide-substituted vinyl arenes that could be converted to *ortho* S-functionalized axially chiral biaryls after post-transformation[27]. Besides these progresses, the development of efficient and general synthetic methods of *ortho* S-substituted axially chiral biaryl structures are still highly desirable. Fascinated by the important function of sulfonyl groups in drug molecules including enhancing potency and binding affinity, modulating solubility, and increasing metabolic stability, we attempted to realize the enantioselective synthesis of potentially valuable *ortho*-sulfonyl-substituted axially chiral biaryl molecules. Pleasingly, the Pd/BaryPhos-catalyzed asymmetric Suzuki-Miyaura cross-coupling was robust to provide a series of

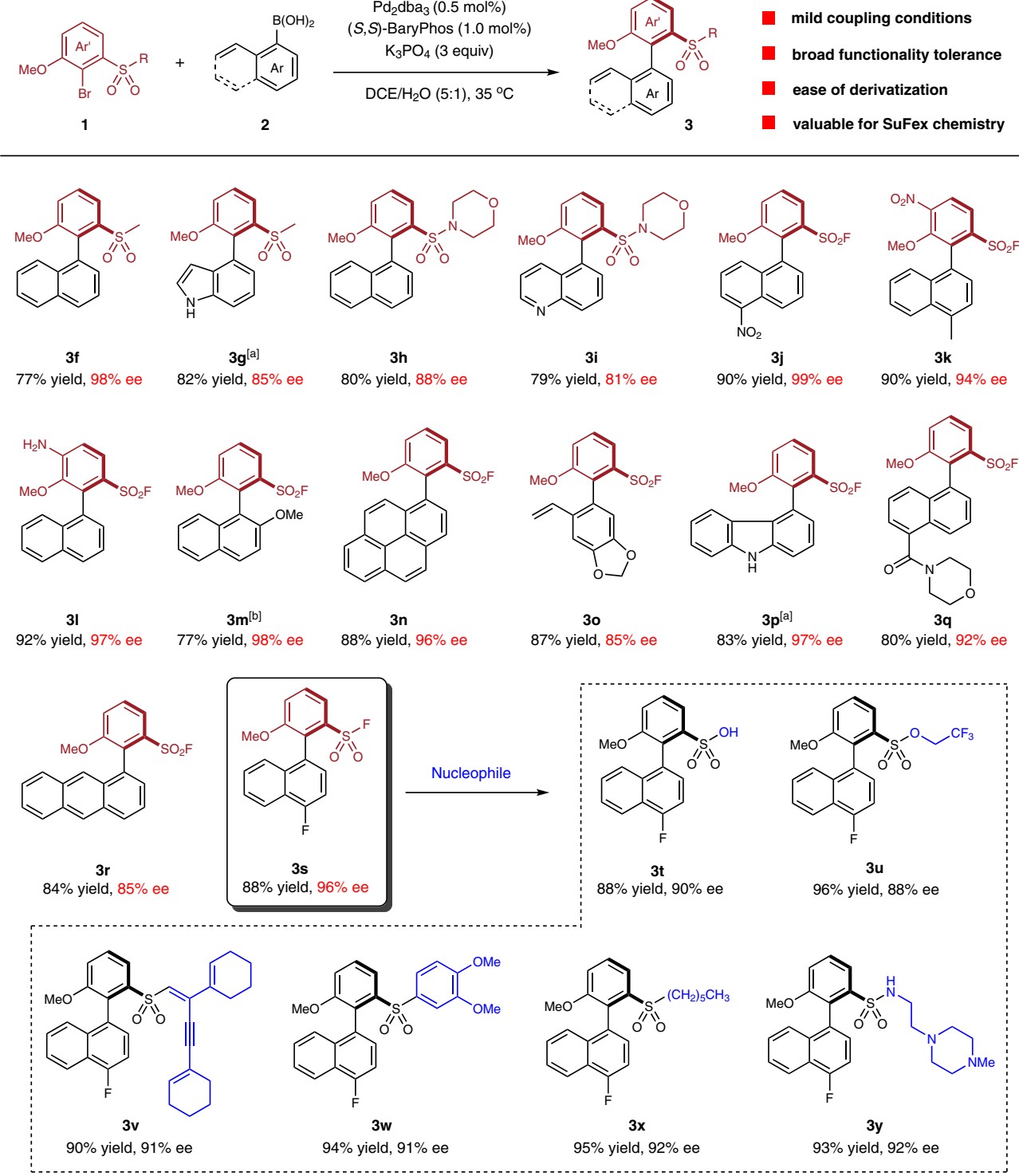

**Fig. 3 | Synthesis of *ortho* sulfonyl-substituted axially chiral biaryls and post-transformation of 3s.** Details of the cross-coupling reactions and derivatization of **3s** were provided in the Supplementary Methods 1.4 and 1.9, respectively; the yields refer to isolated yields; the ee value was determined by HPLC; The absolute configuration of **3n** was determined via its derivative *P*-**17** by X-ray crystallography and the structures of other products were drawn by analogy. [a]The corresponding arylboronic acid pinacol esters **2** were employed. [b]The reaction was conducted at 60 °C. DCE 1,2-dichloroethane.

axially chiral sulfones and sulfonamides in good yields and enantios-electivities at nearly ambient temperature (**3f–i**) (Fig. 3). Unprotected indolyl (**3g**) and quinolyl (**3i**) moieties were well-tolerated. The sulfonyl fluorides are highly desirable compounds in drug discovery and material science due to their 'clickable' characteristic in SuFex chemistry[32,33]. In addition, the propensity to undergo nucleophilic substitution renders them as versatile synthetic intermediates. Aryl

bromides possessing *ortho*-fluoro sulfonyl functionality underwent smooth cross-coupling with substituted 1-naphthylboronic acids to afford axially chiral sulfonyl fluorides (**3j–l**) in excellent yields and enantioselectivities. It was noteworthy that tetra-*ortho*-substituted chiral biaryl **3m** was synthesized with the current coupling protocol at elevated temperature. The coupling reaction was also compatible with various substrate skeletons, such as pyrenyl- (**3n**) and anthracenyl- (**3r**)

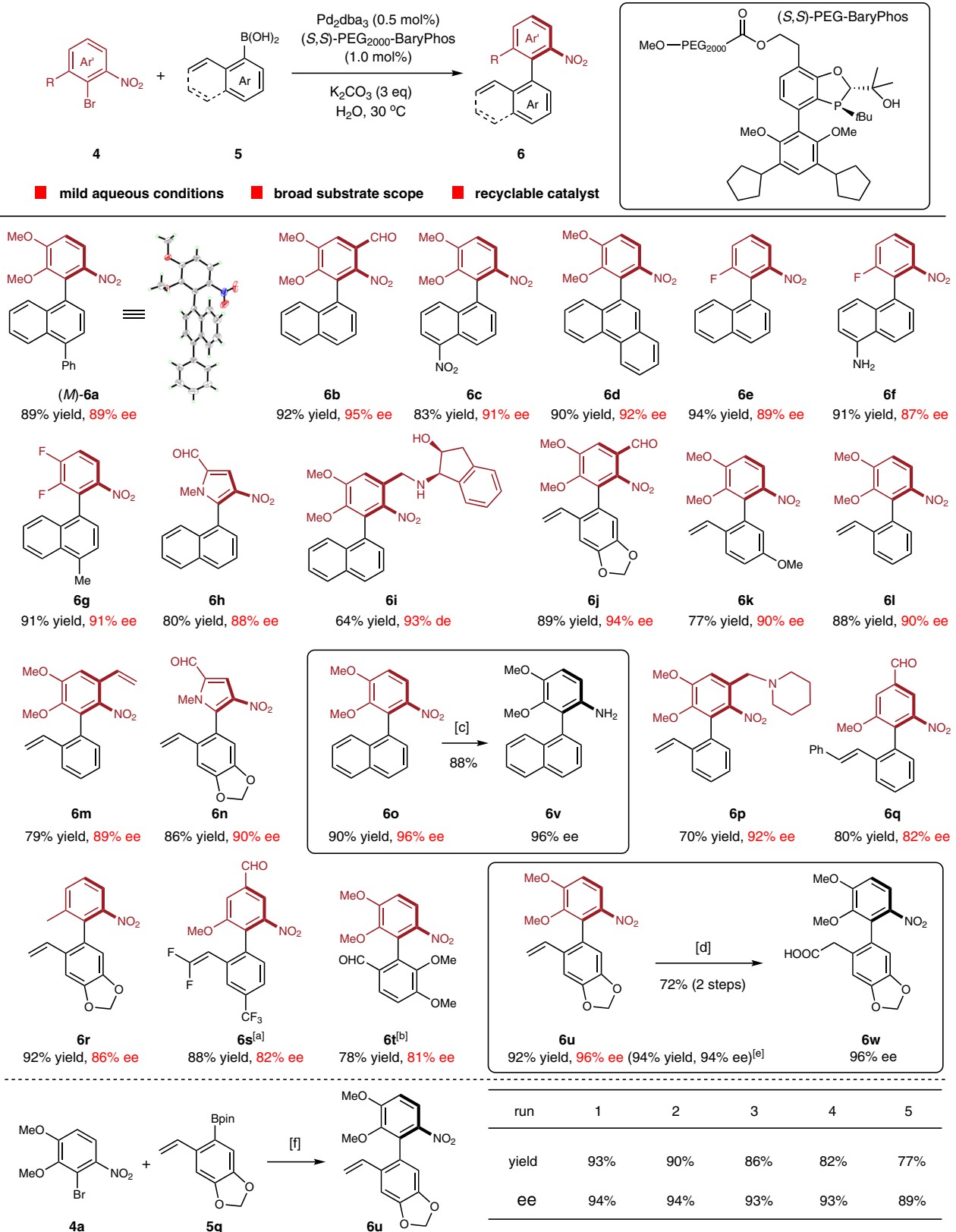

**Fig. 4 | Synthesis of *ortho* nitro-substituted axially chiral biaryls.** Details of the cross-coupling reactions and recycling experiments were provided in the Supplementary Methods 1.4 and 1.5, respectively; yields refer to isolated yields; the *ee* value was determined by HPLC; The absolute configuration of **6a** was determined by X-ray crystallography and the structures of other products were drawn by analogy. [a]The corresponding arylboronic acid pinacol ester was employed. [b]The corresponding potassium aryltrifluoroborate was employed and the reaction was conducted at 70 °C. [c]Reaction condition: Zn, NH₄Cl. [d]Reaction conditions: 1) BH₃ in THF, then NaOH (aq), H₂O₂ (aq); 2) RuCl₃, NaIO₄. [e]Results in parentheses were for reaction using BaryPhos as ligand in Tol/H₂O (5:1) at 30 °C. [f]Reaction condition: Pd₂dba₃ (0.5 mol%), (*S*,*S*)-PEG₁₀₀₀₀-BaryPhos (1.0 mol%), DIPEA (3 eq), H₂O, 35 °C. PEG polyethylene glycol.

**a Existing synthetic approaches (key step)**

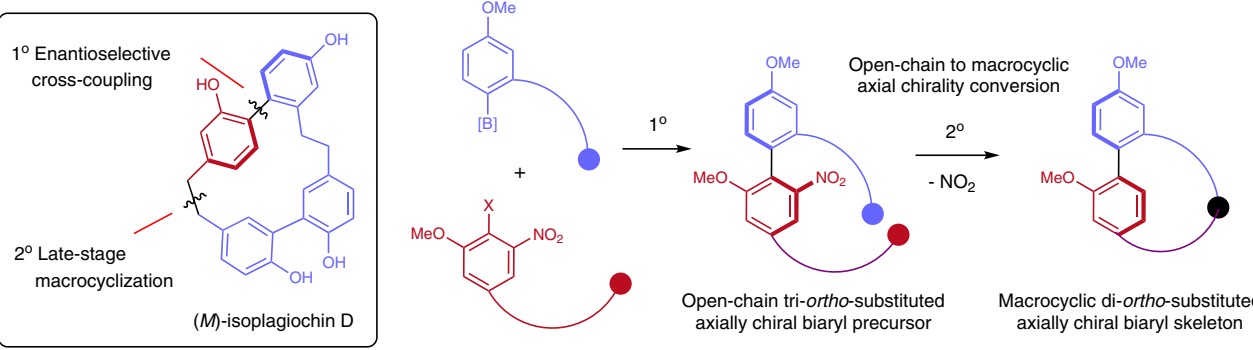

Sulfinyl auxiliary facilitated atropo-diastereoselective Heck reaction (80% yield, 99:1 dr)

(Choppin, Speicher and co-workers)

Atroposelective Heck macrocyclization (22% yield, 37% ee)

(Speicher and co-workers)

key intermediate

Asymmetric macrocyclization of benzyl halide and carbene (22% yield, 93% ee)

(Gu and co-workers)

**b Our synthetic strategy: axial chirality conversion**

1° Enantioselective cross-coupling

2° Late-stage macrocyclization

(*M*)-isoplagiochin D

Open-chain to macrocyclic axial chirality conversion

Open-chain tri-*ortho*-substituted axially chiral biaryl precursor

Macrocyclic di-*ortho*-substituted axially chiral biaryl skeleton

**Fig. 5 | Strategies towards enantioselective synthesis of isoplagiochin D. a** Reported asymmetric synthesis of isoplagiochin D (key step). **b** Our synthetic strategy.

based boronic acids and the one possessing morpholine amide group (**3q**). Besides using naphthalene-type boronic acids, *ortho*-vinyl-substituted phenylboronic acid (**3o**) and carbazolyl boronic ester (**3p**) were suitable substrates and the reactions successfully delivered the corresponding coupling products. The post-functionalization of **3s** led to a library of *ortho* sulfonyl-substituted axially chiral biaryl products. In specific, substitution reaction with hydroxyl and alkoxy nucleophiles provided axially chiral aryl sulfonic acid (**3t**) and 2,2,2-trifluoroethyl arylsulfonate (**3u**) in 88% and 96% yield, respectively. Treatment of **3s** with *n*-hexyllithium or a C(*sp*²) nucleophile provided chiral sulfones **3x** or **3w** in high yields, albeit with a slight loss in enantiomeric excess. The reaction between **3s** and substituted lithium acetylide nucleophile gave **3v** in 90% yield and 91% ee. Presumably, a nucleophilic substitution preceded initially followed by a subsequent Michael addition. Axially chiral sulfonyl fluoride **3s** was also converted to chiral biaryl sulfonamide **3y** in the presence of a lithium amide nucleophile.

## Enantioselective synthesis of *ortho* nitro-substituted axially chiral biaryls

The immobilization of noble metal catalysts offers distinct advantages, such as adjustable solubility of catalyst, ease of catalyst separation, and recycling[34,35]. In the enantioselective synthesis of *ortho*-nitro-substituted axially chiral biaryls, a PEG2000-bound BaryPhos was developed, enabling the asymmetric Suzuki-Miyaura coupling to proceed in water. A serial of axially chiral biaryl products possessing *ortho*-nitro and -methoxy groups were fashioned in high yields and ee's (**6a–d**, **6o**) (Fig. 4). The absolute configuration of **6a** was determined by single crystal X-ray diffraction. Chiral biaryls bearing fluorine atom at the *ortho* position were also synthesized efficiently (**6e–g**). Highly functionalized aryl bromides could also serve as suitable substrates and **6h** containing substituted pyrrole ring was afforded in 80% yield and 88% ee, while **6i** with an aminoindanol pendant was produced in 64% yield

and 93% de. It should be noted that the naphthalene ring was not a necessity for the current enantioselective cross-coupling, as the reaction between aryl bromides and *ortho* vinyl-substituted phenylboronic acids led to chiral biphenyls (**6j–m**, **6p**) and pyrrolyl-phenyl coupling product **6n** with excellent stereochemical fidelity. Functional groups including aldehyde (**6j**, **6n**) and piperidine ring (**6p**) were well tolerated. The coupling reaction was also compatible with 2-styryl- and difluorovinyl-substituted phenyl boronic acid derivatives to give enantioenriched axially biaryl products **6q** and **6s**. Variation of the *ortho*-methoxy unit within aryl bromide to a methyl group (**6r**) had little influence on the reactivity and enantioselectivity of the reaction. With respect to the sterically demanding cross-coupling for the synthesis of tetra-*ortho*-substituted biaryl **6t**, a higher temperature (70 °C) was required to maintain consistent reactivity, while a slightly diminished enantioselectivity was observed. The transformation of these coupling products was straightforward and reduction of nitroarene **6o** by Zn delivered axially chiral amine **6v** in 88% yield and no erosion of ee value was observed. Additionally, chiral biphenyl **6u** underwent a hydroboration-oxidation sequence to give an alcohol intermediate which was converted to axially chiral acid **6w** upon treatment with RuCl3/NaIO4. The recyclability of the catalyst with PEG-supported chiral ligand was demonstrated in the cross-coupling of **4a** and **5g**. Benefited from the sharply different solubility of hydrophilic catalyst in H2O and Et2O, the supported catalyst was reused readily by recycling the aqueous phase, while product **6u** and trace amounts of starting materials was separated by extraction with Et2O. The Pd catalyst supported by PEG10000-BaryPhos was used for four cycles to provide good yields (>80%) and steady enantioselectivity (93-94% ee). Further recycling operations led to a slight decrease in yield and ee of **6u** (77% yield and 89% ee after 4 runs), which was presumed due to Pd leaching or gradual catalyst loss during the recycling process.

**Fig. 6 | Enantioselective synthesis of isoplagiochin D.** 10-step synthesis of (*M*)-isoplagiochin D; detailed experimental information was provided in the Supplementary Fig. 4.

## Enantioselective synthesis of isoplagiochin D

Having established an efficient synthesis of *ortho*-sulfonyl and -nitro-substituted axially chiral biaryls, the application of this general cross-coupling protocol was explored in the asymmetric synthesis of iso-plagiochin D, a cyclophane-type natural product containing an axially chiral di-*ortho*-substituted biphenyl unit embedded in its strained macrocycle. Previous asymmetric synthesis of this macrocyclic bis(bibenzyls) molecule using an asymmetric macrocyclization[36] or Heck reaction[37,38] suffered from either unsatisfactory yields or enantioselectivities (Fig. 5a). We presumed that the axial chirality could be generated prior to the macrocyclic ring closure by using a highly enantioselective cross-coupling, and the obtained tri-*ortho*-substituted axially chiral biaryl intermediate possessing a removable group such as NO₂, could be elaborated as the chiral di-*ortho*-substituted biphenyl moiety (Fig. 5b). Macrocyclic ring closure followed by removal of the nitro group would unambiguously furnish the key chiral di-*ortho*-substituted biphenyl unit. With this concept, our asymmetric synthesis commenced with the preparation of aryl bromide **9** from the Horner-Wadsworth-Emmons reaction between the known aldehyde **8** and phosphonate derivative of **7** (Fig. 6). A subsequent Miyaura borylation converted **9** to boronic ester **10** in 83% yield. In the presence of Pd/(*S,S*)-BaryPhos catalyst, the enantioselective Suzuki-Miyaura cross-coupling of **10** bearing a complex structural skeleton and multi-functionalized **11** furnished chiral biaryl **12** in 70% yield and 90% ee, demonstrating the practicality and robustness of this atroposelective coupling protocol. The tri-*ortho*-substituted biaryl **12** served as the chiral foundation of the synthesis and was transformed to chiral

intermediate **13** under homogeneous hydrogenation using Wilkinson catalyst. It was noteworthy that both the alkene moiety and the more electron-deficient aldehyde group was reduced, which enabled the synthesis of phosphonate **14** after iodination of **13** and subsequent phosphonation. An intramolecular Horner-Wadsworth-Emmons reaction afforded an inseparable *Z/E* isomeric mixture of macrocycle **15** whose alkene group was reduced via catalytic hydrogenation. Under the same condition, the nitro functionality was converted to an amino group which was finally removed by reductive deamination (NaNO₂, H₃PO₂), giving rise to **16** in 46% yield over three steps. A final demethylation afforded isoplagiochin D with *M* configuration in 87% yield. To the best of our knowledge, this 10-step asymmetric synthesis of isoplagiochin D represents an example of building strained axially chiral cyclophane-type natural products from an open-chain chiral biaryl precursor and this efficient strategy should be valuable for the synthesis of macrocycles possessing axially chiral biaryl subunit.

## Photophysical properties investigations

In recent years, the development of chiroptical structures displaying CPL properties has been an attractive area and chromophores based on axially chiral framework have received significant attention[39,40]. The three-dimensional orientation of these molecules endows them with unique photophysical properties at both ground and emitting excited states. Benefited from the convenient synthesis of enantioenriched **3n** through asymmetric cross-coupling, the post-functionalization of this entity led to pyrenyl containing compounds *P*-**17** and *P,P*-**18** (Fig. 7a). The absolute structure of **17** was unambiguously determined by single

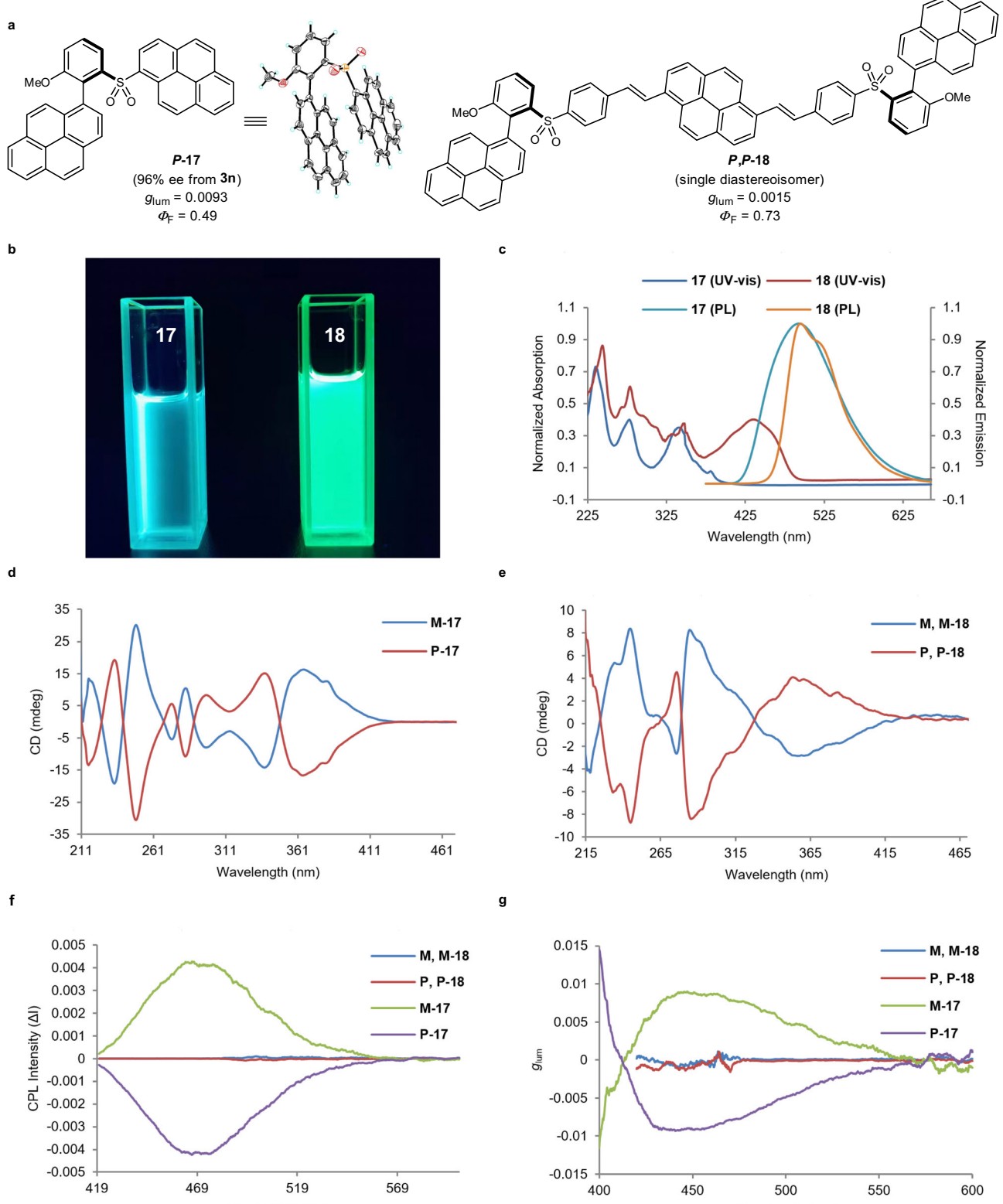

**Fig. 7 | Photophysical properties of chiral compounds 17 and 18. a** Structures of **17** and **18**. **b** Fluorescence images of **17** and **18** in DCM (λ$_{ex}$ = 365 nm). **c** Absorption and fluorescence spectra of **17** and **18** in CH$_2$Cl$_2$ (1.0 × 10$^{-6}$ M), λ$_{ex}$ = 363 nm (for **17**), λ$_{ex}$ = 345 nm (for **18**). **d** CD spectra of *M*-**17** and *P*-**17** in CH$_2$Cl$_2$ (1.0 × 10$^{-5}$ M). **e** CD spectra of *M,M*-**18** and *P,P*-**18** in CH$_2$Cl$_2$ (1.0 × 10$^{-5}$ M). **f** CPL spectra of *M*-**17**, *P*-**17**, *M,M*-**18** and *P,P*-**18** in CH$_2$Cl$_2$ (1.0 × 10$^{-4}$ M). **g** *g*$_{lum}$ values-wavelength curve of *M*-**17**, *P*-**17**, *M,M*-**18** and *P,P*-**18**.

crystal X-ray diffraction and a strong π-π stacking interaction between the two pyrenyl rings was observed in the solid state. Under UV light irradiation (365 nm), the solution of **17** in CH₂Cl₂ exhibited cyan luminescence, while **18** displays bright green luminescence (Fig. 7b). Both compounds show similar UV-vis spectra ranging from 220–350 nm while an additional broad absorption peak was found for **18** at 435 nm due to its extended π system (Fig. 7c). The emission maxima in dilute dichloromethane were recorded at 492 nm and 495 nm for **17** ($\Phi = 0.49$) and **18** ($\Phi = 0.73$), respectively. Strict mirror images were observed for the circular dichroism (CD) spectra of *M*-**17**/*P*-**17** and *M,M*-**18**/*P,P*-**18**, displaying clear cotton effects at several wavelengths (Fig. 7d, e). It was delighting that both pairs of enantiomers *M*-**17**/*P*-**17** and *M,M*-**18**/*P,P*-**18** are CPL-active. In particular, *M*-**17** emits intense positive CPL signals ranging from 420 nm to 550 nm, while *P*-**17** displays a negative signal within the same region (Fig. 7f). In addition, the luminescence dissymmetry factor ($g_{lum}$) of **17** was measured as high as $9.3 \times 10^{-3}$ (Fig. 7g). These chiroptical properties demonstrated the high application potential of related axially chiral molecules in developing new organic chiroptical functional materials.

In this work, a general, efficient and enantioselective synthesis of the *ortho* sulfur- or nitrogen-substituted axially chiral biaryls has been established by a Pd/BaryPhos-catalyzed Suzuki-Miyaura cross-coupling. The versatile protocol shows excellent compatibility to various functional groups and structural features, delivering chiral biaryl structures with *ortho*-sulfonyl groups or with *ortho*-nitro groups at a broad range of molecular diversity and complexity. The development of PEG₁₀₀₀₀-BaryPhos has allowed the asymmetric cross-coupling to proceed in aqueous media and the palladium catalyst recycling for multiple times. Additionally, the method has enabled a concise 10-step asymmetric synthesis of isoplagiochin D as well as the construction of chiroptical molecules with CPL properties, demonstrating its high synthetic values and application potential in natural product synthesis, medicinal chemistry, and material science.

## Methods

### General procedure for asymmetric Suzuki-Miyaura coupling
To a mixture of aryl halide or triflate (0.20 mmol), arylboronic acid or arylboronic ester (0.30 mmol), base (0.60 mmol), Pd₂(dba)₃ (1.0 μmol) and chiral ligand (2.0 μmol, Pd/ligand mol ratio: 1/1) under N₂ was charged degassed solvent. The mixture was stirred at noted temperature for noted time. Ethyl acetate (15 mL) was added and the organic phase was washed sequentially with water (5 mL) and brine (5 mL). The organic layer was separated, dried over Na₂SO₄, filtered and concentrated. The crude enantioenriched chiral biaryl product was purified by silica gel flash column chromatography. A mixture of DCE/H₂O (5:1) was used as solvent and K₃PO₄ as the base for sulfonyl-substituted substrates and the reaction was conducted at 35 °C for 48 h unless otherwise specified. H₂O was used as solvent and K₂CO₃ as the base for nitro-substituted aryl bromide substrates and the reaction was conducted at 30 °C for 12 h unless otherwise specified.

## Data availability
The data that support the findings of this study are available within the paper and its supplementary information files. Source Data of photophysical property studies are provided with this paper. Materials and methods, experimental procedures, characterization data, 1H, 13 C, 19 F NMR spectra and mass spectrometry data are available in the Supplementary Information. The X-ray crystallographic coordinates for structures reported in this study have been deposited at the Cambridge Crystallographic Data Center (CCDC), under deposition numbers CCDC 2130064 (**6a**), and CCDC 2130065 (*P*-**17**). These data can be obtained free of charge from The Cambridge Crystallographic Data Center via www.ccdc.cam.ac.uk/data_request/cif. Source data are provided with this paper.

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

## Acknowledgements
We are grateful to the National Key R&D Program of China 2021YFF0701600 (W.T.), NSFC 22001112 (H.Y.), NSFC 21725205 (W.T.), NSFC 21572246 (W.T.) and Key-Area Research and Development Program of Guangdong Province 2020B010188003 (W.T.) for financial support.

## Author contributions
W.T. directed the project. H.Y. conducted the synthetic experiments and photophysical properties investigations. H.Y. and W.T. wrote the manuscript.

## Competing interests
The authors declare no competing interests.
