## [Peer Review File · Nature Communications]

REVIEWER COMMENTS

Reviewer #1 (Remarks to the Author):

The work presented by Yang and Tang is a significant contribution to the method arsenal for the atroposelective formation of axially chiral biaryls. Whereas the Suzuki-Miyaura cross coupling is recognized today as the most popular method for the construction of “simple” biaryls, its use for the catalytic atropo-enantioselective option is restricted. Atropostable biaryls necessarily possess, in general, at least three more or less bulky ortho-substituents. But this can be crucial for the effectiveness of the cross coupling due to steric hindrance. Elevated temperature may solve the “yield problem”, but on the other hand the atroposelectivity drops down. The BaryPhos ligand with its terhydroxyl group allows a “second interaction” with ortho substituents, especially bearing a strong non covalent hydrogen bond acceptor function. This effect can be considered as a promotor for yield and enantioselectivity as well.

Based on a preliminary work with -CHO substituents, the authors now extend this concept to other functionalities possessing similar electronic properties (see Fig. 2). Among them, the sulfonyl-group (see Fig. 3, ca 14 entries) and the nitro-group (see Fig. 4, ca 22 entries) as ortho-substituents were extensively studied. In the latter case, the PEG-immobilized variant of the BaryPhos ligand was used with respect to sustainability. With a few exceptions, the cross coupling was conducted at 30-35 °C only but with good to best yields and enantioselectivities. For the sulfonyl group, useful post-functionalization options were shown (Fig. 3, below). The enantioselective synthesis of isoplagiochin D was presented as an application of the ortho-nitro group coupling finally leading to cyclophane type natural product. At the end, some photophysical experiments/results were presented for the more complex compounds 17/18.

General remarks:

1) All chiral biaryls were defined by “R” or “S”. Why not by aR/aS or better M/P notification as used finally for the isoplagiochin D enantiomer as well as for 17/18?

2) Although the mode of enantiocontrol of the Pd-BaryPhos-catalytic system was previously discussed, a general prediction rule should be given for (S,S)-or (R,R)-BaryPhos. The absolute configuration was assigned in all cases without any comment. The verification of the absolute configuration was given only for the “chiroptical compounds” 17/18.

3) Did the authors compare for any example the effectiveness of the “free” versus immobilized version of the BaryPhos ligand? The latter one seems to be not readily available (compare the SuppInf part).

4) On page 9, references are given for alternative asymmetric syntheses of isoplagiochin D. At least the basic concepts should be given, comparing with the authors concept in Fig. 5a. In this context, the authors state that their synthesis “represents the first example of building strained axially chiral

cyclophane type natural products from an open chain chiral biaryl precursor...". On the other hand and to my opinion, this seems to be the less exciting way to do this.

Special remarks:

page 2, line 4: add a reference behind "drug candidates"

page 2, line 7: delete "extremely"

page 3, line 1: add "are" mostly...

page 4, Figure 1a: The header is somewhat confusing as achiral molecules are presented in column 3

page 4, Figure 1b: Here as well as in the main text not mentioned is the sulfinyl group as ortho-functionality. This group also allows post-functionalization, and, in addition, atropo-diastereoselective coupling (extensive work by the Colobert group).

page 6, line 5 and others: Please do not use "excellent" enantioselectivities (some examples only are in the range of 80 %)

General Remarks for the Supporting Information document:

1) All experiments are well documented, all compounds are sufficiently characterized including enantiomer analyses comparing with a racemic probe.

2) For many starting materials the origin is not documented (literature or supplier).

Special remarks:

page S10, first scheme: same formulas for S16/S15

page S41/42: compound 6k: wrong chromatogram?

page S64/S65: Is the compound really pure? compare shoulder in the tailing peak

Reviewer #3 (Remarks to the Author):

In this manuscript, the authors studied the atroposelective Suzuki reaction catalyzed by Pd/BaryPhos using a broad scope of sterically hindered aryl halides, affording the C-C axially chiral biaryl products bearing diverse ortho substituents around the chiral axis such as nitro, sulfonyl, ester, alkenyl, and CN groups. The reaction proceeded well using a chiral ligand that they have previously described. From the synthetic point of view, the system development is not that original and there is barely new concept by replacing previous groups into the N and S group at the ortho position. Is it a big deal by introducing sulfonyl and nitro groups? The authors seem to indicate that CHO, ester and nitro groups were rarely studied during creation of chiral axis, but this statement seems to be limited

to the case of asymmetric Suzuki reactions. These groups can be well embedded when it comes to other asymmetric coupling reactions. The authors seem to go too far by claiming this originality.

On the other hand, the reviewer has noted that the development of PEG10000-BaryPhos has allowed the reaction to proceed in water and the palladium catalyst is recyclable for several times. This method also enabled a 10-step asymmetric synthesis of isoplagiochin D and the construction of chiroptical molecules. Photophysical properties of selected products have also been provided. From the conceptual weakness but with the synthetic advances, I leave the decision to the editor.

The following revisions are suggested prior to resubmission to any journal.

- (1) Can the Pd catalyst supported by PEG-BaryPhos catalyze the o-sulfonyl-substituted substrates? Will it also allow for multiple recycling of the catalyst when utilizing PEG with a larger molecular weight?
- (2) The ee value of isoplagiochin D should be given. Does the ee decay after multistep reactions?
- (3) "The absoluteX-ray diffraction [17]" What's the relationship between the structure of compound 17 and reference 17?
- (4) Delete the "Sulfur- or Nitrogen-Substituted" in the title. Why is a big deal?
- (5) In Figure 2, authors indicated 2nd interaction(s) between the P and the Aryl groups. Are you sure that they are trans to each other? This is pure speculation unless they provided evidence. It should be deleted. Don't oversell yourself by putting forward speculated concepts.
- (6) The ee of the product 3d is lower than that of the rest products. Is it related to the relatively low racemization barrier? This barrier should be measured and provided to the main text.

In this paper, Yang and Tang described an efficient method for enantioselective synthesis of ortho sulfur- or nitrogen-substituted axially chiral biaryls via a chiral Suzuki-Miyaura coupling protocol powered by a chiral monophosphorus ligand-BaryPhos in excellent yields and enantioselectivities. This was another landmark effort by the authors on asymmetric Suzuki-Miyaura coupling particularly for the synthesis of chiral biaryl structures with ortho-sulfonyl groups. The high compatibility with various functional groups was shown. The usefulness of this protocol was exquisitely displayed with a concise 10-step asymmetric synthesis of isoplagiochin D as well as the construction of CPL active chiroptical molecules whose preparations were mostly based on chemical resolution. Furthermore, the immobilization of BaryPhos on PEG support and recycling of the chiral palladium catalyst was successfully demonstrated. Overall, the authors have presented a complete story on asymmetric Suzuki-Miyaura coupling with high effectiveness, practicality, and synthetic applications. The work should attract readers from synthetic organic chemistry, drug discovery and materials science. The work is recommended for publications if the following comments can be addressed:

- 1) The authors presented the asymmetric Suzuki-Miyaura protocol applicable to a variety of functionalized axially chiral biaryls. It will be great if an empirical model of substrate features can be provided for readers and users.
- 2) It seems that this method has addressed the most issues for construction of axially chiral biaryls. It would be better that the authors mention the remaining challenges in the area.
- 3) Figure 3c: In the structure of biaryls shown in the box, the structure of R group was already clearly shown. It was not necessary to indicate "R = NO₂" and "R = SO₂R", which are suggested to be shown under the structure of

bromide.

4) The configuration of compound 6 was determined by X-ray structure analysis. However, the determination of compound 3 was not mentioned. How was the configuration of compound 3 determined?

5) For the synthesis of compound 6, the PEG-bound BaryPhos was utilized to apply this reaction in water. Is the immobilization of BarylPhos on PES also applicable to the synthesis of compound 3?

6) Is it possible to give a plausible explanation for the configuration selectivity of the formed biaryls?

REVIEWER COMMENTS

Reviewer #1 (Remarks to the Author):

The work presented by Yang and Tang is a significant contribution to the method arsenal for the atroposelective formation of axially chiral biaryls. Whereas the Suzuki-Miyaura cross coupling is recognized today as the most popular method for the construction of “simple” biaryls, its use for the catalytic atropo-enantioselective option is restricted. Atropostable biaryls necessarily possess, in general, at least three more or less bulky ortho-substituents. But this can be crucial for the effectiveness of the cross coupling due to steric hindrance. Elevated temperature may solve the “yield problem”, but on the other hand the atroposelectivity drops down. The BaryPhos ligand with its terhydroxyl group allows a “second interaction” with ortho substituents, especially bearing a strong non covalent hydrogen bond acceptor function. This effect can be considered as a promotor for yield and enantioselectivity as well. Based on a preliminary work with -CHO substituents, the authors now extend this concept to other functionalities possessing similar electronic properties (see Fig. 2). Among them, the sulfonyl-group (see Fig. 3, ca 14 entries) and the nitro-group (see Fig. 4, ca 22 entries) as ortho-substituents were extensively studied. In the latter case, the PEG-immobilized variant of the BaryPhos ligand was used with respect to sustainability. With a few exceptions, the cross coupling was conducted at 30-35 °C only but with good to best yields and enantioselectivities. For the sulfonyl group, useful post-functionalization options were shown (Fig. 3, below). The enantioselective synthesis of isoplagiochin D was presented as an application of the ortho-nitro group coupling finally leading to cyclophane type natural product. At the end, some photophysical experiments/results were presented for the more complex compounds 17/18.

General remarks:

1) All chiral biaryls were defined by “R” or “S”. Why not by aR/aS or better M/P

notification as used finally for the isoplagiochin D enantiomer as well as for 17/18?

Response: Thanks for the reviewer's suggestion. For consistence and accuracy, we adopted the M/P notification across the whole manuscript and made changes to the content and figures (Fig. 4 and 5).

2) Although the mode of enantiocontrol of the Pd-BaryPhos-catalytic system was previously discussed, a general prediction rule should be given for (S,S)- or (R,R)-BaryPhos. The absolute configuration was assigned in all cases without any comment. The verification of the absolute configuration was given only for the "chiroptical compounds" 17/18.

Response: Thanks for the reviewer's comment.

Absolute configuration assignment:

In order to assign the absolute configuration of products, we managed to obtain X-ray crystal structures of *ortho* nitro-substituted biaryl **6a** (in Fig. 4) and *ortho* sulfonyl-substituted biaryl **17** which was synthesized from **3n** (in Fig. 3). The synthesis of **17** from **3n** is stereospecific and the procedure is included in Supplementary Information S66. Accordingly, the absolute structures of the two classes of coupling products were drawn by analogy. Information on absolute configuration assignment was noted in the corresponding Figure Legends (Fig. 3 and 4).

Prediction rule and stereochemical model:

Careful analysis of the configuration relationship between chiral ligand and compounds **6a** and **17** (or **3n**) indicated that the current mode of enantiocontrol with respect to the non-covalent interaction and steric effect was in accordance with the one we previously discussed. Therefore, a stereochemical model was proposed to demonstrate the mode of catalysis and prediction role of configuration. To maintain preciseness, we also restricted the structure features of substrates (we have studied to date) suitable for this model. This information

has been discreetly included in the Supplementary Information (Supplementary Figure 1, Page S56) and is shown below.

In general, BaryPhos is a suitable ligand for asymmetric Suzuki-Miyaura cross-coupling with aryl halide substrates containing ortho functionalities, such as nitro, sulfonyl, cyano, aldehyde, ester, and alkenyl groups. The aryl boronic acid/ester coupling partners include substituted 1-naphthaleneboronic acid/esters or related structures, ortho alkenyl-substituted aryl boronic acids/esters.

a) Suitable substrates:

b) Stereochemical model:

A stereochemical mode was proposed to explain the enantiocontrol of the coupling reaction and to predict the configuration of the products. Based on our previous report and the coupling reactions studied in this work, it is believed that a hydrogen bonding between the tertiary alcohol of BaryPhos and the ortho functionality of aryl halide dominates the orientation of Ar¹. A C-H/ π interaction between cyclopentyl group of ligand and Ar² exists presumably to set the conformation of this aryl coupling partner. Reductive elimination of the above intermediate delivers the chiral biaryl product with observed configuration.

3) Did the authors compare for any example the effectiveness of the “free”

versus immobilized version of the BaryPhos ligand? The latter one seems to be not readily available (compare the SupplInf part).

Response: Thanks for the reviewer's comment. The immobilized BaryPhos was not readily available or studied previously. This is the first time that a PEG supported BaryPhos (P-chiral ligand) has been synthesized and investigated in asymmetric cross-coupling. We indeed compared the catalytic performance of 'free' and 'supported' BaryPhos in the synthesis of biaryl compound **6u**. The result was included in Fig. 4, as well as in Supplementary Information (Supplementary Table 1, Page S53).

	With PEG ₂₀₀₀ -BaryPhos	With BaryPhos		92% yield	94% yield	
	96% ee	94% ee	

The reaction using MeO-PEG₂₀₀₀-BaryPhos provided **6u** in 92% yield and 96% ee. The parallel reaction with 'free' BaryPhos led to comparable result (94% yield, 94% ee). This indicates that both ligands are effective for the current cross-coupling reactions. Probably, the tethering of PEG support and BaryPhos via a two carbon alkyl chain is uninfluential to the electronic property of the ligand (related to reactivity). In addition, the PEG chain is far away from the P-chiral center (as well as the Pd metal center), which is considered to pose little influence on enantioselectivity.

4) On page 9, references are given for alternative asymmetric syntheses of isoplagiochin D. At least the basic concepts should be given, comparing with the authors concept in Fig. 5a. In this context, the authors state that their synthesis "represents the first example of building strained axially chiral cyclophane type natural products from an open chain chiral biaryl precursor...". On the other hand and to my opinion, this seems to be the less exciting way to do this.

Response: We agree with the reviewer that the synthetic concepts of reported alternative asymmetric syntheses of isoplagiochin D should be provided. Accordingly, we revised Fig. 5 and the added information was contained in Fig. 5a.

In specific, Choppin, Speicher and co-workers furnished the key macrocyclisation and axial chirality through a chiral sulfinyl auxiliary facilitated diastereoselective Heck coupling (15 steps from known starting materials). Speicher and co-workers also employed an atroposelective Heck macrocyclization (key step: 22% yield, 37% ee) using chiral Pd catalyst in the synthesis of (M)-isoplagiochin D (16 steps, LLS). A atroposelective macrocyclisation between benzyl halide and carbene was developed for the synthesis of this natural product (key step: 22% yield, 93% ee, 11 overall steps).

a Existing synthetic approaches (key step)

b Our synthetic strategy: axial chirality conversion

As shown above, the reported syntheses features a common concept that the axial chirality is generated in the macrocyclisation step. However, relatively low yields were usually obtained for this challenging step. We turned to an alternative approach to build an open chain axially chiral biaryl foundation first, and then conduct the macrocyclisation to forge the ring-strained bisbibenzylis skeleton. Although most reactions seems to be 'routine', we believe that this synthetic sequence indeed improves the synthetic efficiency and provides a reliable method to other axially chiral cyclophane structures.

Special remarks:

page 2, line 4: add a reference behind “drug candidates”

Response: Two references have been added behind ‘drug candidates’. Please see ref. [4] LaPlante, S. R. et al. Accessing atropisomer axial chirality in drug discovery and development. *J. Med. Chem.* **54**, 7005-7022 (2011); [5] Clayden, Moran, W. J., Edwards, P. J. & LaPlante, S. R. The challenge of atropisomerism in drug discovery. *Angew. Chem. Int. Ed.* **48**, 6398-6401 (2009).

page 2, line 7: delete “extremely”

Response: The word ‘extremely’ has been deleted.

page 3, line 1: add “are” mostly...

Response: We added ‘are’ in the sentence.

page 4, Figure 1a: The header is somewhat confusing as achiral molecules are presented in column 3

Response: Thanks for the reviewer’s comment. We are aware of the presence of misleading achiral molecules in Fig 1a. Our primary consideration of including two achiral molecules in column 3 is to show the prevalence and importance of aryl sulfonyl-substituted structures in approved drugs and candidates. During our collaboration with medicinal chemists, we learned that axially chiral sulfonyl-substituted arenes are promising pharmacophores but usually avoided in drug discovery due to their synthetic challenge. Therefore, the pharmacological profiles of these molecules is still unknown and herein we developed a reliable synthetic method in this study.

To avoid confusion, we revised the header for Fig. 1a as ‘Axially chiral molecules with ortho N- or S-functionalization, prevalence of achiral aryl sulfonyl-containing drugs and potential pharmacological value of axially chiral sulfonyl substituted arenes’.

page 4, Figure 1b: Here as well as in the main text not mentioned is the sulfinyl group as ortho-functionality. This group also allows post-functionalization, and, in addition, atropo-diastereoselective coupling (extensive work by the Colobert group).

Response: Thanks for the reviewer's comment. The asymmetric synthesis of ortho S-substituted axially chiral biaryl structures has been rather limited to date. The sulfoxide-directed asymmetric C-H activation strategy developed by Colobert and co-workers represent an effective and inspirational approach to ortho sulfinyl-substituted chiral biaryl compounds. Additionally, Zhao and co-workers reported an electrophilic carbothiolation of alkynes for the synthesis of chiral sulfide-substituted vinyl arenes that could be converted to ortho S-functionalized axially chiral biaryls.

We included these progresses (with references) in the 'Enantioselective synthesis of ortho sulfonyl-substituted axially chiral biaryls' section of the main text.

Added content: 'The ortho S-substituted axially chiral biaryl compounds possess significant application potential in drug discovery and material science. However, their asymmetric synthesis remains a challenging task. Pioneering work from Colobert and co-workers enabled the efficient synthesis of ortho sulfinyl-substituted axially chiral biaryls via a sulfoxide-directed asymmetric C-H functionalization strategy. Zhao and co-workers realized an electrophilic carbothiolation of alkynes for the synthesis of chiral sulfide-substituted vinyl arenes that could be converted to ortho S-functionalized axially chiral biaryls after post-transformation. Besides these progresses, the development of efficient and general synthetic methods of ortho S-substituted axially chiral biaryl structures are still highly desirable.'

page 6, line 5 and others: Please do not use "excellent" enantioselectivities (some examples only are in the range of 80 %)

Response: Thanks for the reviewer's comment. We deleted 'excellent' in page 6, line 5, double-checked and addressed similar phrases across the manuscript in order to be objective and accurate.

General Remarks for the Supporting Information document:

1) All experiments are well documented, all compounds are sufficiently characterized including enantiomer analyses comparing with a racemic probe.

Response: Thanks for the reviewer's comment and we made corrections related to the rest remarks below.

2) For many starting materials the origin is not documented (literature or supplier).

Response: Thanks for the reviewer's comment. We double-checked the Supplementary Information, included the supplier for commercial materials in the 'General Information' section and added references for reported known compounds.

Special remarks:

page S10, first scheme: same formulas for S16/S15

Response: The structure of starting material S16 has been corrected.

page S41/42: compound 6k: wrong chromatogram?

Response: Thanks for the reviewer's comment. The chromatogram is correct but the original data was from another HPLC condition for this compound which led to a slight overlap between the peaks of the two enantiomers. We have updated the HPLC data which are consistent with the current chromatogram.

page S64/S65: Is the compound really pure? compare shoulder in the tailing peak

Response: Thanks for the reviewer's comment. The compound (**3x**) was

further purified and the HPLC analysis method was optimized. The HPLC data and traces have been updated in Supplementary Information (S68/S69).

Reviewer #2 (Remarks to the Author):

In this paper, Yang and Tang described an efficient method for enantioselective synthesis of ortho sulfur- or nitrogen-substituted axially chiral biaryls via a chiral Suzuki-Miyaura coupling protocol powered by a chiral monophosphorus ligand-BaryPhos in excellent yields and enantioselectivities. This was another landmark effort by the authors on asymmetric Suzuki-Miyaura coupling particularly for the synthesis of chiral biaryl structures with ortho-sulfonyl groups. The high compatibility with various functional groups was shown. The usefulness of this protocol was exquisitely displayed with a concise 10-step asymmetric synthesis of isoplagiochin D as well as the construction of CPL active chiroptical molecules whose preparations were mostly based on chemical resolution. Furthermore, the immobilization of BaryPhos on PEG support and recycling of the chiral palladium catalyst was successfully demonstrated. Overall, the authors have presented a complete story on asymmetric Suzuki-Miyaura coupling with high effectiveness, practicality, and synthetic applications. The work should attract readers from synthetic organic chemistry, drug discovery and materials science. The work is recommended for publications if the following comments can be addressed:

1) The authors presented the asymmetric Suzuki-Miyaura protocol applicable to a variety of functionalized axially chiral biaryls. It will be great if an empirical model of substrate features can be provided for readers and users.

Response: Thanks for the reviewer's comment. A stereochemical model was proposed and shown below based on our results to demonstrate the mode of catalysis. In general, BaryPhos is a suitable ligand for asymmetric Suzuki-Miyaura cross-coupling with aryl halide substrates containing ortho functionalities, such as nitro, sulfonyl, cyano, aldehyde, ester, and alkenyl groups. The aryl boronic acid/ester coupling partners include substituted 1-

naphthaleneboronic acid/esters or related structures, ortho alkenyl-substituted aryl boronic acids/esters.

a) Suitable substrates:

b) Stereochemical model:

We are also aware that there might be exceptions for this empirical predication. Therefore, we discreetly included this model and description in the Supplementary Information (Supplementary Figure 1, Page S56).

2) It seems that this method has addressed the most issues for construction of axially chiral biaryls. It would be better that the authors mention the remaining challenges in the area.

Response: Thanks for the reviewer's comment. In the current study, we provided an efficient cross-coupling for the synthesis of ortho nitro- and sulfonyl-functionalized axially chiral biaryls and applied this method to the synthesis of isoplagiochin D and optical materials. However, we are aware of the limit of the current method and other general challenges for asymmetric Suzuki-Miyaura coupling. For example, there are still numerous important axially chiral biaryl natural products which have not been able to be synthesized with reported coupling methods to date. In addition, only a limited number of available asymmetric Suzuki-Miyaura couplings are suitable for sterically

demanding and highly functionalized substrates. Therefore, highly robust and versatile coupling methods are still lacking to address these challenges.

We added this information in the Introduction section of the manuscript and rephrased the original description of the main text as below:

‘Despite the advancement, the employment of enantioselective Suzuki-Miyaura coupling in the synthesis of axially chiral natural products and high value-added molecules remains a challenging task and only a limited number of application have been reported to date.¹⁶⁻²⁰ In addition, synthetic and medicinal chemists nowadays still suffer from the situation of few readily available, practical and robust enantioselective cross-coupling protocols for the efficient synthesis of highly functionalized bioactive biaryl intermediates with axial chirality.’

3) Figure 1c: In the structure of biaryls shown in the box, the structure of R group was already clearly shown. It was not necessary to indicate “R = NO₂” and “R = SO₂R”, which are suggested to be shown under the structure of bromide.

Response: Thanks for the reviewer’s suggestion. We adjusted the position of ‘R = NO₂ or SO₂R’ in Fig. 1c to be shown below the aryl bromide structure.

4) The configuration of compound 6 was determined by X-ray structure analysis. However, the determination of compound 3 was not mentioned. How was the configuration of compound 3 determined?

Response: The *ortho* sulfonyl-substituted biaryl **17** in Fig. 6 was determined by X-ray structure analysis and this compound was actually synthesized from **3n** in Fig. 3 (The synthesis of **17** from **3n** was included in Supplementary Information S66). Accordingly, the absolute structures of coupling product **3** were drawn by analogy with that of **17**. Information on absolute configuration assignment was noted in the corresponding Figure Legends (Fig. 3 and 4).

5) For the synthesis of compound 6, the PEG-bound BaryPhos was utilized to apply this reaction in water. Is the immobilization of BaryPhos on PEG also applicable to the synthesis of compound 3?

Response: We did investigate the performance of PEG-BaryPhos in the synthesis of ortho sulfonyl-substituted chiral biaryls (**3**). The Pd/PEG-BaryPhos catalyzed coupling reactions provided comparable yields and ees to the ones using unbound 'free' BaryPhos. Nevertheless, unsatisfactory results were obtained in recycling experiments due to a high degree of Pd leaching and gradual catalyst loss.

Based on our observation and analysis, one reason was considered to be responsible for this undesirable result. Due to the increased steric bulkiness and coordinating ability (to Pd) of the sulfonyl group in comparison with nitro group, the coupling reactions with sulfonyl-substituted substrates generally require longer reaction time (around 48 h) and relatively higher temperature (35 °C or above) than those with nitro-substituted aryl bromides (detailed reaction conditions were included in the Supplementary Information). Consequently, a trace amount of Pd black was observed at the end of reactions due to partial dissociation of ligand and the subsequent 2nd run gave a decreased yield.

6) Is it possible to give a plausible explanation for the configuration selectivity of the formed biaryls?

Response: Thanks for the reviewer's comment. A stereochemical mode was proposed and shown below to explain the enantiocontrol of the coupling reaction and to predict the configuration of the products. Based on our previous report and the coupling reactions studied in this work, it is believed that a hydrogen bonding between the tertiary alcohol of BaryPhos and the ortho functionality of aryl halide dominates the orientation of Ar¹. A C-H/ π interaction between cyclopentyl group of ligand and Ar² exists presumably to set the conformation of this aryl coupling partner. Reduction elimination of the above intermediate delivers the chiral biaryl product with observed configuration.

Stereochemical model:

This proposed model and description have been included in the Supplementary Information (Supplementary Figure 1, Page S56).

Reviewer #3 (Remarks to the Author):

In this manuscript, the authors studied the atroposelective Suzuki reaction catalyzed by Pd/BaryPhos using a broad scope of sterically hindered aryl halides, affording the C-C axially chiral biaryl products bearing diverse ortho substituents around the chiral axis such as nitro, sulfonyl, ester, alkenyl, and CN groups. The reaction proceeded well using a chiral ligand that they have previously described. From the synthetic point of view, the system development is not that original and there is barely new concept by replacing previous groups into the N and S group at the ortho position. Is it a big deal by introducing sulfonyl and nitro groups?

Response: We appreciate the reviewer's comment and evaluation on our study. We conceived this project basing on the following considerations and this study is expected to be a breakthrough and contribution in asymmetric cross-coupling.

The importance and synthetic challenges of ortho N- or S-substituted axially chiral biaryls

The ortho nitro- or sulfonyl-substituted axially chiral biaryls possess high application value in synthetic and medicinal chemistry. However, their efficient synthesis has been rarely studied by asymmetric cross-coupling approach (The asymmetric synthesis of these molecules by other strategies has also been

limited). From our collaborations with medicinal and process chemists, we learned that the ortho N-, S, and halogen-substituted chiral biaryls are of great value to drug discovery but usually avoided due to their synthetic challenge. Therefore, we were dedicated to addressing this challenge and provided a synthetic solution to these chiral structures.

Further development of asymmetric Suzuki-Miyaura cross-coupling

The asymmetric Suzuki-Miyaura coupling has become an indispensable transformation in synthetic chemistry. However, most reported work has been focused on purely methodology development with simply functionalized substrates. Substrates possessing ortho nitro- or sulfonyl-functionalities have been seldom investigated or poorly tolerated in these studies (as shown in Supplementary Information S55 and Supplementary Table 1 about ligand comparison in the synthesis of ortho nitro-substituted axially chiral biaryls). It is pleasing to develop an efficient coupling protocol compatible with ortho N-/S-substitution and we are confident that the expansion of the boundary of asymmetric Suzuki-Miyaura cross-coupling will be beneficial to asymmetric synthesis.

The authors seem to indicate that CHO, ester and nitro groups were rarely studied during creation of chiral axis, but this statement seems to be limited to the case of asymmetric Suzuki reactions. These groups can be well embedded when it comes to other asymmetric coupling reactions. The authors seem to go too far by claiming this originality.

Response: Thanks for the reviewer's comment. We agree with the reviewer that other asymmetric coupling reactions (as well as other synthetic methods) are available for the construction of ortho CHO, ester and nitro substituted axially chiral biaryl structures. Nevertheless, the efficient synthesis of these molecules has been limited via asymmetric Suzuki-Miyaura coupling which is one of the most straightforward and practical synthetic method in medicinal and

process chemistry, as well as large scale production. With respect to asymmetric Suzuki-Miyaura coupling, this study represents a significant breakthrough and progress.

For clarity and avoiding misleading descriptions, we restricted the originality only to asymmetric Suzuki-Miyaura cross-coupling and replace 'cross-coupling' with 'Suzuki-Miyaura cross-coupling' in the Introduction Section of the manuscript.

On the other hand, the reviewer has noted that the development of PEG10000-BaryPhos has allowed the reaction to proceed in water and the palladium catalyst is recyclable for several times. This method also enabled a 10-step asymmetric synthesis of isoplagiochin D and the construction of chiroptical molecules. Photophysical properties of selected products have also been provided. From the conceptual weakness but with the synthetic advances, I leave the decision to the editor.

Response: We appreciate the encouragement on the synthetic advances of this coupling protocol from reviewer and editor. We are aware of the fact that the chiral ligand BaryPhos and the Suzuki-Miyaura coupling in the study are not new, which the reviewer considers as conceptual weakness. Nevertheless, we would like to address the following strength and features of this study for your consideration:

In-depth exploration of the value of BaryPhos-mediated asymmetric cross-coupling

While our previous work focused mainly on the discovery of BaryPhos and its primary application in Suzuki-Miyaura cross-coupling, which was only the tip of the iceberg, this study further exploited the versatile application of BaryPhos in the synthesis of ortho sulfur or nitrogen substituted axially chiral biaryls that are most relevant to drug discovery and natural product chemistry, as documented in this manuscript. This work timely delivers the message when synthetic and

medicinal chemists suffer the situation of few readily available, practical and robust chiral ligand for asymmetric cross-coupling. The advent and commercialization of BaryPhos provides the solution that many medicinal and process chemists are hopeful. Therefore, the in-depth exploration of the application value of this robust ligand and practical coupling protocol is believed to be highly important and beneficial.

Demonstration of enantioselective coupling in natural product synthesis

Although dramatic progress has been achieved in catalytic asymmetric Suzuki-Miyaura cross-coupling, most reported work has been focused on mainly methodology development and only limited applications have been realized in natural product synthesis to date. The key issues remain lack of prominent chiral catalysts/ligands and poor functional group compatibility of the existing coupling protocols. Taking advantage of the BaryPhos-facilitated asymmetric Suzuki-Miyaura cross-coupling, we managed to furnish the asymmetric synthesis of isoplagiochin D within 10 steps from readily available starting materials. This represent the shortest synthesis of this natural product and the strategy was conceived to be a general and practical method for the synthesis of cyclophane-type structures containing axially chiral biaryl moieties. This study is also considered to be a contribution to natural product synthesis via enantioselective cross-coupling approach.

The following revisions are suggested prior to resubmission to any journal.

(1) Can the Pd catalyst supported by PEG-BaryPhos catalyze the o-sulfonyl-substituted substrates? Will it also allow for multiple recycling of the catalyst when utilizing PEG with a larger molecular weight?

Response: Thanks for the reviewer's comment on PEG-BaryPhos and the corresponding coupling reactions.

Recycling experiments in the synthesis of ortho sulfonyl-substituted chiral biaryls

We did investigate the recycling of Pd catalyst supported by PEG-BaryPhos in the synthesis of ortho sulfonyl-substituted chiral biaryls. Although Pd/PEG-BaryPhos catalyzed coupling reactions provided comparable yields and ees to the ones using 'free' BaryPhos, unsatisfactory results were obtained in recycling experiments due to a high degree of Pd leaching and gradual catalyst loss. Based on our observation and analysis, one reason was considered to be responsible for this undesirable result. Due to the increased steric bulkiness and coordinating ability (to Pd) of the sulfonyl group in comparison with nitro group, the coupling reactions with sulfonyl-substituted substrates generally require longer reaction time (around 48 h) and relatively higher temperature (35 °C or above) than those with nitro-substituted aryl bromides (detailed reaction conditions were included in the Supplementary Information). Consequently, a trace amount of Pd black was observed at the end of reactions due to partial dissociation of ligand and the subsequent 2nd run gave a decreased yield.

Immobilization of ligand utilizing PEG with a larger molecular weight

We agree with the reviewer that the utilization of PEG support with a larger molecular weight is usually beneficial for recyclability due to their stability, ease of recycling and decreased gradual catalyst loss. In our study, we indeed prepared PEG₂₀₀₀₀-BaryPhos, along with PEG₂₀₀₀-, PEG₅₀₀₀- and PEG₁₀₀₀₀-BaryPhos. As shown below, the immobilization of BaryPhos on PEG₂₀₀₀ or PEG₅₀₀₀ resulted in noticeable catalyst loss during two-phase separation process and reduced reactivity was observed in subsequent runs. When supported ligand bearing larger molecular weight (PEG₂₀₀₀₀-BaryPhos) was used, it was challenging to conduct the two-phase recycling process since gel was formed in the aqueous layer. To date, PEG₁₀₀₀₀-BaryPhos provided the optimal catalytic performance for this asymmetric cross-coupling in aqueous media and enabled catalyst recycling for several times.

	With PEG ₂₀₀₀ -BaryPhos or PEG ₅₀₀₀ -BaryPhos	PEG ₁₀₀₀₀ -BaryPhos	PEG ₂₀₀₀₀ -BaryPhos
--	--	--------------------------------	--------------------------------

 6u	Noticeable catalyst loss (soluble in both phases during two-phase separation process)	Optimal molecular weight (good recyclability and catalytic performance)	Challenging recycling process (two-phase separation was not possible due to gel formation)
--	---	---	--

According to our study and literature, the recyclability of supported catalysts is largely dependent on reaction conditions (such as solvent and temperature), properties of substrate/product and supported catalyst (solubility, stability, etc.). We are aware that the current result is still not perfect or general for practical application. We will continue with our effort to improve the immobilization of chiral ligands for reliable catalysis in the future.

(2) The ee value of isoplagiochin D should be given. Does the ee decay after multistep reactions?

Response: The ee value of isoplagiochin D has been included in Fig. 5 of the manuscript and the HPLC traces and separation conditions were added in the Supplementary Information S77. No erosion of ee was observed after multistep synthesis from the key Suzuki-Miyaura coupling reaction, which was attributed to the mild reaction conditions (<50 °C) for the whole synthetic sequence.

(3) “The absoluteX-ray diffraction [17]” What’s the relationship between the structure of compound 17 and reference 17?

Response: We really appreciate the reviewer’s comment and apologize for this irrelevant citation which has been deleted in the main text.

(4) Delete the “Sulfur- or Nitrogen-Substituted” in the title. Why is a big deal?

Response: Thanks for the reviewer’s comment. The asymmetric Suzuki-Miyaura coupling has become an indispensable transformation in synthetic chemistry. Significant progress has been made in the past two decades, as demonstrated by tremendous publications in this area. Nevertheless, there has been no report on the asymmetric synthesis of ortho-sulfonyl-substituted axially

chiral biaryl molecules and very rare studies of ortho-nitro-substituted axially chiral biaryl synthesis by asymmetric Suzuki-Miyaura cross-coupling. Considering that the ortho N or S substituted axially chiral biaryls are highly valuable structures and the biological profiles of these compounds are in active investigation in industry, we conceived this project and developed an efficient and reliable synthetic protocol. We believe that the work should attract a broad readership from medicinal chemists as well as process chemists. To highlight this specific progress and breakthrough, we would like to keep the key words 'Sulfur- or Nitrogen-Substituted' in the Title if possible.

(5) In Figure 2, authors indicated 2nd interaction(s) between the P and the Aryl groups. Are you sure that they are trans to each other? This is pure speculation unless they provided evidence. It should be deleted. Don't oversell yourself by putting forward speculated concepts.

Response: Thanks for the reviewer's suggestion. We are aware of the inappropriate drawing of the model. This was proposed as a cartoon image to show the 2nd interactions between chiral ligand and both substrates based on current result and our previous study, and was not designed to demonstrate the configuration of the real Pd complex during the reaction. However, it is indeed confusing to represent ligand with 'P'. Therefore, we modified this cartoon image and the ligand was labeled as 'L'.

(6) The ee of the product 3d is lower than that of the rest products. Is it related to the relatively low racemization barrier? This barrier should be measured and provided to the main text.

Response: Thanks for the reviewer's comment. The configurational stability of product **3d** and **3e** was studied. As demonstrated by the plots of ee as a function of time, both compounds were configurationally stable at the reaction temperatures (30 °C for **3d** and 35 °C for **3e**), as the ee values generally maintained constant within 2 days. The thermal racemization barriers of **3d** ($\Delta G^\ddagger = 27.6$ kcal/mol, 60 °C) and **3e** ($\Delta G^\ddagger = 29.0$ kcal/mol, 60 °C) were also calculated. The racemization energy barrier has been provided in the manuscript (Figure 2) and detailed information has been included in Supplementary Information (Supplementary Figure 2, Page S57).

To explain the enantiocontrol of the coupling reaction, a stereochemical mode was proposed (as shown below and also included in the Supplementary Information S56 as other reviewers requested). Based on the results in this work and our previous report, it is believed that a hydrogen bonding between the tertiary alcohol of BaryPhos and the ortho functionality of aryl halide dominates the orientation of Ar¹. A C-H/ π interaction between cyclopentyl group of ligand and Ar² exists presumably to set the conformation of this aryl coupling partner. Reduction elimination of the above intermediate delivers the chiral biaryl product with observed configuration.

Stereochemical model:

While both **3d** and **3e** were considered to be configurationally stable under reaction conditions, the enantioselectivity for **3d** (89% ee) did not compete with that for **3e** (97% ee), which might be due to the weaker interaction between the cyano group of substrate and tertiary alcohol moiety of the ligand (than the case with sulfonyl substituted substrates), thus leading to decreased ee value of **3d**.

REVIEWERS' COMMENTS

Reviewer #1 (Remarks to the Author):

The authors carefully checked and responded to my (and other) reviewer remarks. The answers were extensive as well as rich in content with respect to scientific discussion. Where needed, all corrections and completions were done in the revised manuscript as well as Supp. Inf.

To my opinion, this manuscript can now be published with no further revision.

Reviewer #2 (Remarks to the Author):

The manuscript has been improved satisfactory with all the points from the Reviewers being fully addressed. Herein, I would like to recommend its publication on Nature Communications.

< In comments to the Editorial office, Reviewer 1 noted that Reviewer 3's prior concerns were adequately met in this round of review. >